# Socioeconomic Factors Influence the Spatial and Temporal Distribution of Blue–Green Infrastructure Demand: A Case of Nanjing City

**DOI:** 10.3390/ijerph20053979

**Published:** 2023-02-23

**Authors:** Haixia Zhao, Binjie Gu, Jinding Fan, Junqi Wang, Liancong Luo

**Affiliations:** 1Nanjing Institute of Geography and Limnology, Chinese Academy of Sciences, Nanjing 210008, China; 2Nanjing School, University of Chinese Academy of Sciences, Nanjing 211135, China; 3School of Environmental Science and Engineering, Suzhou University of Science and Technology, Suzhou 215009, China; 4Institute for Ecological Research and Pollution Control of Plateau Lakes, School of Ecology and Environmental Science, Yunnan University, Kunming 650504, China

**Keywords:** blue–green infrastructure, measurement of demand, spatial and temporal differences, influencing factors, Nanjing

## Abstract

Blue–green infrastructure provides a variety of ecosystem services and is becoming an increasingly vital part of urban ecosystem protection. It is an ecological facility for ecological conservation and environmental protection, and a foundation for realizing people’s needs for a better life. This study selects indicators from four dimensions: social, economic, environmental, and ecological, and the demand for blue–green infrastructure is assessed comprehensively. The results show that: (1) the demand for blue–green infrastructure varies spatially with the development of the city; (2) the total demand for blue–green infrastructure in Nanjing from 2000 to 2020 shows a pattern of “high in the center and low in the periphery”; (3) the level of economic development, urban spatial pattern, and decision management orientation have different degrees of influence on the demand for blue–green infrastructure, with the urban spatial pattern having the greatest impact. Therefore, in the future, blue–green infrastructure should be optimized by taking into account the spatial characteristics of demand in Nanjing.

## 1. Introduction

Blue–green infrastructure (BGI) typically refers to an interconnected network of multifunctional green spaces and blue spaces that are strategically planned and managed to increase water retention and connectivity between natural areas to provide a range of ecological, social and economic benefits [1,2,3]. Accommodating the acceleration of urbanization, the expansion of urban construction land has caused the fragmentation of the natural landscape, which also leads to multi-scale environmental and ecological problems in metropolitan areas. Urban heat island effect, air pollution, and inner-city waterlogging have become the crises and challenges faced by most cities in the world [4,5]. BGI can provide various ecosystem services, such as climate regulation, air purification, carbon sink increase, biodiversity protection, and aesthetic and cultural services. BGI is considered a nature-based solution that eliminates the adverse effects of urbanization [6,7].

Demand for BGI refers to the needs of nature and people for the social, economic, ecological, and environmental functions that BGI provides. The supply of and demand for BGI highlight its ecological and social functions, respectively [8]. Usually, the functions of BGI increases with its scales, but the imbalance of local BGI supply and demand matching often generates local service function surplus or deficiency. The sustainable development of the local societal economy, along with the continuing advancement of eco-civilization, has led governments and regulators to pay more attention to the various functions of BGI in city planning and construction. Under the construction method of “cutting the needle, leaving blank and adding green,” the urban BGI is gradually improving, and the supply capacity is greatly improved, which plays a vital role in promoting regional welfare and economic development, improving people’s well-being and improving the local ecological environment [9,10,11]. It is, however, lacking foresight in planning and constructing BGI commensurate with social, economic, ecological, and environmental demand. This results in the fragmented development of the central urban area and concentrated construction of the suburbs [12]. The social function of BGI cannot be fully realized, which seriously restricts its overall efficiency. In the context of natural resource constraints, scientifically measuring the demand for BGI in urban development can clarify the driving mechanism for changes in the spatial and temporal pattern of demand and provide a scientific basis for the construction of BGI to promote sustainable urban development.

BGI is often seen as a way to meet the demand of local residents for social, economic, ecological, and environmental benefits. As residents seek higher quality habitats, the demand for BGI increases accordingly [8,13]. At present, BGI demand can be measured in three ways: (i) based on ecosystem services; (ii) based on indicator construction; and (iii) based on economic methods, such as questionnaires reflecting willingness to pay. There has been research showing that the demand for BGI research starts from a measurement of ecosystem services based on the theory of supply and demand [14,15]. Some scholars constructed an evaluation system for demand that comprehensively considers population density, land development intensity, and socio-economic indicators, such as GDP and the night lighting index [16]. Some scholars conduct evaluations of the demand for food, water supply, carbon sequestration, oxygen release demand, air purification requirements, high-temperature regulation requirements, recreational demand, and many other aspects, respectively [17,18]. Several studies have used questionnaires to measure urban residents’ willingness to pay and their payment for BGI [19]. In existing studies on the construction and evaluation of indicator systems, there are many methods for constructing a demand evaluation index system, which is mainly carried out from two dimensions: the economic and social perspectives [20]. The economic perspective reflects the benefits of residents directly and indirectly accessing and spending on BGI. The social perspective reflects residents’ social aspirations, preferences, and demand for BGI. It achieves this by selecting indicators, such as social demand, consumption, preference, perception, or economic metrics, that ignore urban residents’ demands for ecological and environmental aspects [21]. There are few existing studies that consider the four aspects of economy, society, ecology, and environment at the same time. Along with the development of urban social economy and the improvement of living standards of urban residents, it is necessary to reform the urban BGI evaluation system, so that it takes into account social, economic, ecological, and environmental multi-dimensional urban BGI requirements to meet the increasing demand for a healthy scenic environment. Reform should include air purification, rain and flood regulation, temperature control, and maintaining biodiversity in the evaluation index [22,23,24,25].

Demand for BGI is influenced by many factors. National and local policies and regulations determine protection efforts and the distribution of demand within the city. The increase in residents’ incomes and living standards promotes the increased demand for BGI, and historical and cultural accumulation, population distribution, and major social events also promote the increased demand for BGI in the region [26,27,28]. In addition, public awareness and participation and the living environment of different urban areas under the perspective of environmental justice also influence the demand of urban residents [29]. In conclusion, the distribution of demand for BGI is mainly influenced by the level of socio-economic development, but relevant studies are still relatively limited. With cities’ development, the demand for BGI shows significant regional differences. Human activities caused the mismatch between the supply and demand of BGI, which is the principal influencing factor for the geographical demand discrepancy. However, there are other factors, such as economic development level, urban spatial layout, and decision management orientation, etc. Among them, the level of economic development includes GDP, industrial layout, fixed asset investment, etc., which primarily represents the mode and level of regional socio-economic development. The urban spatial layout mainly consists of the urban layout system, which determines the protection intensity and the internal demand distribution of the city. Decision management involves laws and regulations, policies, management methods, etc. It is the decisions made by government departments in the planning, construction, and management of BGI. Guidance and management of human activities directly or indirectly affect the demand for BGI.

The main contribution of this paper lies in two points. First, the index system of BGI demand is constructed by considering four dimensions: social, economic, ecological, and environmental, and the demand of people and nature for each dimension of BGI is evaluated comprehensively. Second, it initially explores the influencing factors affecting the distribution of BGI demand and enriches the understanding of BGI demand. Nanjing, capital city of Jiangsu Province, China, is studied as an example to measure the demand for BGI from the four dimensions. We also analyze the temporal and spatial variation patterns. This study identifies the influencing factors of demand and provides a theoretical basis for the rational planning and construction of BGI.

## 2. Materials and Methods

### 2.1. Study Area Overview

Nanjing, located at 118°22′ E~119°14′ E, 31°14′ N~32°37′ N, is the capital of Jiangsu Province and the political, economic, and cultural center (Figure 1). Nanjing is an important central city in the Yangtze River Delta region and an imperative node city at the intersection of the Yangtze River Economic Belt and the Coastal Economic Belt. The climate is classified as the north subtropical monsoon climate, with an average annual precipitation of 1090.4 mm, an average annual temperature of 13~22 °C, and prevailing northeast and southeast winds. Nanjing positions north–south long, east–west narrow. In the territory of Nanjing, the terrain contains hillock, plain, low-lying ground, marshes, low mountains, and hills as a geomorphic complex. By 2020, the city had a total area of 6587 km^2^, including 868.28 km^2^ of built-up area. It has jurisdiction over 11 municipal districts and one state-level new district (Jiangbei New Area), with a permanent population of 9,319,700 and an urbanization rate of 86.8%.

### 2.2. Demand Evaluation Indicators

According to the benefit system of BGI, this study constructs an index evaluation system from the four dimensions of society, economy, environment, and ecology. It determines the index weight through the entropy weight method, thereby comprehensively assessing the demand for BGI.

**The society.** From an urban social perspective, urban residents, participants in the real estate industry, and government workers influence BGI [30]. As residents are the main body of urban society, BGI has a profound impact on the health and well-being of urban residents. Providing recreational and fitness venues for urban residents to help improve their physical and mental health is an important guarantee for sustainable urban development and the health and well-being of residents [31]. With the deepening of urbanization, urban development has many characteristics, such as high population density and high development intensity. On the one hand, excessive urban population density will lead to water resource stress, traffic congestion, environmental pollution, etc. Refined BGI will help to relieve the problems caused by a dense population. On the other hand, with the increase in building density, residents are more inclined to exercise in the natural environment, such as parks and green spaces. This is to relieve pressure and meet their spiritual needs. Therefore, population density and building density are chosen to characterize the demand for BGI in the social dimension.

**The economy.** Economic variables influence the management and demand distribution of BGI [32]. On the one hand, due to the limited land resources and economic differences in urban built-up areas, the contradiction between supply and demand of BGI is particularly prominent in urban built-up areas, mainly due to the increased demand for BGI caused by the agglomeration of urban economic activities. In the process of urban development and growth, the maintenance of ecosystems should be considered. The concept of ecological and green economy is injected into the economic development, and high-quality urban development replaces the rough urban development for sustainable urban development. Therefore, land development intensity is considered as an economic dimension characteristic of BGI demand. On the other hand, urban night lights represent the economic development level of cities [33,34]. The stronger the night lights are, the more frequent human activities are, and the higher the level of urban development. It needs to be supported and enhanced by more BGI. Therefore, night lights are also used to characterize the BGI demand of the economic dimension.

**The ecology.** BGI has the function of carbon sequestration and habitat maintenance [35,36]. The ability of BGI to sequester carbon and release O_2_ is achieved through photosynthesis of daytime vegetation, and its size, vegetation type, and growth condition all have a significant impact on its carbon sequestration and oxygen release function [37]. Greenhouse gas emissions exacerbate climate change, and urban BGI plays an important role in mitigating climate change. Carbon emission indicates that the region needs BGI to exert its carbon sequestration function to achieve carbon neutrality, so carbon emission is taken as an indicator of BGI demand in an ecological dimension. Concurrently, BGI provides a good habitat for urban organisms and plays an important role in maintaining the biodiversity of the city. However, urban development and construction split the urban natural corridor, resulting in fragmented local landscape patterns and weakened the service function of BGI [38]. BGI can realize the connection between the quality of the human living environment and healthy wild animals [39]. The fragmentation index can indicate the urban ecological threat, so as to represent the ecological dimension of BGI demand.

**The environment.** BGI allows climate and pollution regulation. The heat island effect is a major environmental problem plaguing urban development, formed by poor heat diffusion caused by buildings and roads in cities. Vegetation can reduce the surrounding air temperature and create a high-temperature regulation service through shading, transpiration, and evaporation. The evapotranspiration process of BGI is effective in absorbing heat from the atmosphere and also has a certain shading function, which is significant in mitigating urban heat island problems [40]. Therefore, the surface temperature is selected to measure the demand level of urban residents for the air temperature regulation function of BGI. In addition, the environmental regulation of BGI is also reflected in its air purification function. Green vegetation can adsorb and degrade specific air pollutants from the air [22]. Therefore, surface temperature and PM_2.5_ concentration were selected as environmental requirements for characterizing the BGI.

#### 2.2.1. Source and Processing of Indicator Data

(1)Population density

Population grid data were obtained from the Resource and Environmental Science and Data Center of the Chinese Academy of Sciences (https://www.resdc.cn/, accessed on 30 September 2021). Data from 2000 and 2010 were selected for this study. The spatial distribution of the population in Nanjing in 2020 was calculated based on 2019 mobile phone signaling data and the 7th Nanjing Population Census data. The specific formula is as follows:(1)$$R_{i}=POP_{j}\times \left(q_{i}/Q_{j}\right)$$

In the formula: $R_{i}$ is the population of the i, $POP_{j}$ is the seventh census data of the j, $q_{i}$ is the mobile phone signaling value of the i, and $Q_{j}$ is the total mobile phone signaling value of the j.

(2)Building density

The data on floor area in 2000 and 2010 were obtained from the Nanjing Real Estate Yearbook, and the floor area data in 2020 were obtained from the WISE (Water Information Service Explorer) information service platform (http://www.zkyq-tech.cn/, accessed on 10 October 2021). Considering the accessibility, precision, and scale of the data, a 1km grid was used to calculate the building density in the area. The specific formula is as follows:(2)$$R_{i}^{t}=m_{t}{\left(0.5y_{it}+0.5r_{it}\right)}_{g}/S_{i}$$

In the formula: $R_{i}^{t}$ is the building density of the i in t year, t stands for 2000, 2010, and 2020, $S_{i}$ is the construction land area of the i, $m_{t}$ is the building area in t year, and $y_{it}$ is year t the night light index of the i, $r_{it}$ is the population of the i in year t, and g is the normalized index.

(3)Night light

Existing studies have shown that night light change is affected by urbanization and industrialization, which is directly related to economic indicators [41,42]. The higher the night light value, the better the economic development and the greater the demand for BGI, and vice versa. The data were obtained from the National Center for Environmental Information of the United States (https://ngdc.noaa.gov/, accessed on 1 March 2022). The three phases of night light data were processed by mutual correction, continuity correction, image synthesis, denoising, and other processes.

(4)Intensity of land development

The higher the land development intensity, the better the land’s economic benefit and the higher its land price. As a result, land has a relatively low economic value. The specific formula is as follows:(3)$$L_{i}=B_{i}/S_{i}$$

In the formula: $L_{i}$ is the land development intensity index of the i, $B_{i}$ is the area of construction land in the i, and $S_{i}$ is the total area of the i.

(5)Carbon emissions

Based on the amount and type of Nanjing energy use, this study focuses on electricity, coal, coke, natural gas, gasoline, liquefied petroleum gas, and high carbon emissions energy. It also accounts for district energy use. In this study, space parameters of night lights and demographic data are calculated based on night lights, population, and the entire social power consumption structure. Results provide a reasonable determination of the spatial distribution of energy consumption in Nanjing. By combining relevant references, the carbon emission coefficient of the main energy sources was calculated (Table 1) [43]. The energy data were obtained from the environmental statistics of Jiangsu Province and the Statistical Yearbook of Nanjing City. The specific formula is as follows:(4)$$\left\{\begin{array}{c} \beta_{n}=c_{n}/S_{n},\gamma_{n}=b_{n}/S_{n}\\ C_{i}={\left(\beta y_{in}+\gamma r_{in}\right)}_{g}\sum m_{jn}\alpha_{j} \end{array}\right.$$

In the formula: $\beta_{n}$ is the night light parameter in the n year, $\gamma_{n}$ is the population parameter in the n year, $c_{n}$ is the industrial power consumption in the n year, $S_{n}$ is the total social power consumption in the n year, $b_{n}$ is the urban and rural area in the n year residential electricity consumption, $C_{i}$ is carbon emissions of i in year n, $y_{in}$ is the night light index of i in year n, $r_{in}$ is the population of i in year n, *g* is the normalized index, $m_{jn}$ is the energy consumption of the j in the n year, and $\alpha_{j}$ is the carbon emission coefficient of the j of energy.

(6)Fragmentation

The fragmentation index of BGI is calculated by ArcGIS10.5, and the fragmentation calculation formula is as follows:(5)$$P_{i}=S_{GIi}/M_{GIi}$$

In the formula: $P_{i}$ is the fragmentation index of the i, $M_{GIi}$ is the area of BGI in the i, and $S_{GIi}$ is the number of BGIs in the i.

(7)Surface temperature

Nanjing’s land surface temperature data from 2000 to 2020 were retrieved using the atmospheric correction method, Landsat series data of geographical data space cloud (https://www.gscloud.cn/, accessed on 11 October 2021), and Envi software (version 5.3.1, Exelis Visual Information Solutions, Inc., Boulder, CO, USA). In order to ensure comparability, remote sensing satellite image data in June 2000, May 2010, and May 2020 were used. This was performed to obtain that the average temperature of Nanjing in 2000, 2010, and 2020 was 26.2 °C, 25.3 °C, and 25.4 °C, respectively. This was in agreement with the actual temperature.

(8)PM_2.5_

PM_2.5_ data were gathered from NASA’s earth observation data and information in the system of social and economic data and application center (https://sedac.ciesin.columbia.edu/, accessed on 28 February 2022), 1998–2020, the global annual PM_2.5_ network data.

#### 2.2.2. Weight Determination

ArcGIS software (version 10.5, Environmental Systems Research Institute, Inc. (Esri), Redlands, CA, USA) was used to construct 1 km × 1 km fishing nets in Nanjing, and the values of each index in the region were extracted. Considering that the nature and units of each indicator have strong differences and are difficult to be unified, in order to better evaluate the demand for BGI and its spatial distribution in Nanjing, the weights of each indicator are determined by the entropy value method after normalizing the data of individual indicators. Under data normalization, the weight of each index was determined by the entropy method, as shown in Table 2. The entropy weight method is based on the degree of variation of the index values of each indicator to determine the index weights, which is an objective assignment method to avoid the bias brought by human factors. Compared with those subjective weight methods, it is more accurate and objective, and can better explain the obtained results.

### 2.3. Influencing Factors of Blue–Green Infrastructure Demand

A regression analysis was used to measure the impact of different factors on BGI demand. The variables are measured in different units of measurement in a multiple regression analysis; therefore, we used the standardized regression coefficients to determine which of the factors have a greater effect on the BGI demand according to the equation [44]:(6)$$\beta^{*}=\beta \times \frac{\sigma_{x}}{\sigma_{y}}$$
where $\beta^{*}$ is the standardized regression coefficient, β is the regression coefficient resulting from the linear regression, and $\sigma_{x}$ and $\sigma_{y}$ are the estimated standard deviations of x and y, respectively.

According to the previous description, we believe that three factors, namely the level of economic development, urban spatial pattern and decision management orientation, influence the distribution of BGI demand, as shown in Table 3. The level of economic development is characterized by GDP and industrial park layout, and denoted by 1 and 0 according to whether it is an industrial park site or not. The spatial layout of the city is assigned different scores according to the functional level of the city, with 3 points for the main city, 2 points for the secondary city, 1 point for the new city, and 0 points for the rest of the city. Decision management orientation is characterized by the use of new parks or not. Since parks are public in China and their new construction is often influenced by government decisions, new parks are used to replace the influence of policy factors on BGI demand, with new parks assigned a value of 1 and the rest 0. The above indicators are assigned to the fishnet and analyzed in stata16.

## 3. Results

### 3.1. Changes in the Spatial and Temporal Pattern of Demand

The study measures and analyzes the BGI demand in Nanjing from the aspects of social demand, economic demand, ecological demand, and environmental demand.

#### 3.1.1. Social Demand

Social demand for BGI increased first and then decreased with the increase in population size and building density in Nanjing. This increased from 0.0112 in 2000 to 0.0273 in 2010 and 0.0248 in 2020. The data indicate an overall increase in social demand. Variations in social demand in different districts are affected by differences in building density. The social demand in Jianye district and Yuhuatai district in the central city decreased first and then increased. The data show that the increase rate in Xuanwu District and Qinhuai District was 252% and 429%, respectively. Due to growth in development opportunities since 2000, the three southern districts have seen a rise in population and construction land area. This has led an increase in building density and social demand.

In terms of space, the social demand for BGI in Nanjing is mainly concentrated in the central urban area, along the S8 metro line, the north of Jiangning, Lukou street, the political and cultural center of Lishui and Gaochun and the other regions, which are greatly influenced by the dense population in these areas (Figure 2). The social demand in 2000 clearly had an industrial and economic orientation. The area with the highest demand was located at the junction of Getang Street, Pancheng Street, and Dachang Street. These three streets have been the most developed industrial areas in Nanjing since the founding of the People’s Republic of China. In 2010, the high social demand shifted to the city’s center and the middle of the Pukou District. The social demand in the middle part of Jiangning District is obviously greater than that in the surrounding areas. This is mainly because of the dense building density. The distribution of high demand will become widespread with the further improvement of urban function layout. In 2020, the service industry dominated the city center, which is still the area with the largest social demand. In addition, the concentrated population in the developed manufacturing areas, such as Yuhua Economic Development Zone, Xiongzhou Street, and Getang, have relatively high social demand.

#### 3.1.2. Economic Demand

Under the background of high-quality urban development, land use is more intensive and efficient, and the high-value region of nighttime light is more concentrated. The economic demand for BGI in Nanjing has been reduced from 0.0946 to 0.0260, with a reduction rate of 72.52%. The economic demand for the most developed urban areas declined yearly, and the decline rate reached more than 60%. The suburbs and outer suburbs have relatively lagged economic development, and the economic demand has drastically diminished from 2000 to 2010. The economic demand keeps growing with the increase in investment in fixed assets.

Regarding space, high demand aggregation is mainly distributed in the central urban area, along the S8 metro line, the north of Jiangning, Lukou Street, the political and cultural center of Lishui District and Gaochun District, etc. (Figure 3). Core economic demand was concentrated in the city’s central area, where the junction of Gulou, Xuanwu, Jianye, and Qinhuai districts was located in 2000. In 2010, the city center’s core economic demand was transferred to Hexi’s new city center. Many companies have settled in the Jianye district. Its high-demand area continues to increase with the revitalization of the old town and the establishment of Hexi’s new city center. Xinjiekou, the city center, will remain in high demand until 2020, as will Nanjing South Railway Station, the other core of the new high-speed rail city.

#### 3.1.3. Ecological Demand

Ecological demand for BGI in Nanjing rose from 0.0160 in 2000 to 0.0319 in 2020, with a growth rate of 99.4%. Data indicate that the expansion of urban construction land and continuous population growth directly cause the reduction and fragmentation of ecological space under the backdrop of the strengthened eco-environment protection policies. The ecological demand of all regions increased, and the ecological demand of Gaochun District, Xuanwu District, and Gulou District increased the most, reaching 68.96%, 63.11%, and 58.57%, respectively. This is due to the improvement of people’s lives. Qixia District, Jianye District, Lishui District, and Luhe District increased next to the previous three districts, with an increasing rate of 40%. The BGI network center in Jiangning District, with high development and construction intensity, however, has been continuously eroded. This district also has a high degree of BGI fragmentation, the largest population, and more developed industries, which leads to the largest increase in ecological demand.

In terms of space, the ecological demand core of spatial pattern lay in the Gulou District from 2000 to 2020 (Figure 4). In 2000, landscape fragmentation outside the central city was further exacerbated by urban expansion and development along the river. In the central city of Jiangnan and the new main city of Jiangbei, this resulted in scattered large ecological demands. In 2010, the ecological demand in Gulou District decreased. Ecological demand in developing manufacturing areas, such as Nanjing Chemical Industry Park and southeast of Longpao new city, was considerable. The northern part of the Yangtze River midstream (Nanjing section) had a large number of manufacturing enterprises settled down by 2020 due to the diversion of manufacturing from the central urban area. The high ecological demand area has grown steadily. In addition, the Nanjing High-speed Railway Hub Economic Zone boasts significant social and economic benefits. This area has experienced a steady increase in carbon emissions, fragmentation, and ecological demand.

#### 3.1.4. Environmental Demand

Since 2000, the Nanjing city government has promulgated regulations on the prevention and management of water, air, and soil pollution. The level of environmental governance has continuously improved. The average environmental demand was generally maintained between 0.0113 and 0.0114. The environmental demand of the Yuhuatai district remained unchanged, while the environmental demand of other districts in the central city increased slightly, and the increment level was below 0.0005. Pukou District and Luhe District have established many manufacturing enterprises in the central urban area, leading to fluctuating environmental demand. The eco-environment of Jiangning District, Lishui District, and Gaochun District has improved, and environmental demand has decreased.

In terms of space, the spatial distribution of environmental demand in Nanjing changed significantly from 2000 to 2020 (Figure 5). The high-demand area shifted from south to north due to the spatial distribution of PM_2.5_. In 2000, the high environmental demand in Nanjing was mainly concentrated along the coastline of the Yangtze River, in the middle and southern regions of Jiangning, the eastern and southern regions of Lishui, and the middle and western regions of Gaochun District. In 2010, the high environmental demand area was transferred to northern Luhe, eastern Pukou, Nanjing High-speed Railway Hub Economic Zone, Nanjing Airport Hub Economic Zone, and other regions. The environmental demand in Pukou District is increasing rapidly due to the pollution remedy needs of industrial enterprises. In 2020, the environmental demand in Lishui District and Gaochun District decreased significantly. However, the high environmental demand areas were concentrated in Luhe District, the central urban area, and the northern region of Jiangning.

#### 3.1.5. Overall Demand

In general, the social and ecological demand for BGI in Nanjing increased from 2000 to 2020, while economic demand declined, and environmental demand increased slightly. At the same time, the overall demand for BGI in Nanjing decreased yearly. There is a noticeable variation in the circle layer distribution that forms a decreasing core area, mainly located in Xuanwu District, Gulou District, Qinhuai District, Jianye District, and Qixia District, a decreasing suburban area mainly located in Pukou District, Yuhuatai District, Jiangning District and Lishui District, and an outer suburban decreasing area mainly located in Luhe District and Gaochun District. The continuous improvement of the industrial development level of various economic development zones in Nanjing has promoted industrial development in the suburbs and a high-quality labor force gathering in the industrial parks. Therefore, the total demand for BGI in Jiangning District, Pukou District, and Lishui District has been increasing in recent years. The total demand for BGI in Gaochun District and Luhe District is relatively small compared to urban areas.

From 2000 to 2020, the total demand for BGI in Nanjing showed a pattern of “high in the center and low in the periphery,” in which BGI was mainly distributed in the central urban area (Figure 6). The core is at the junction of Xuanwu, Jianye, Gulou, and Qinhuai. It is concentrated in the political and cultural centers along the S8 metro line, the northern regions of Jiangning, Lukou Street, Meishan Street, Gaochun, and Lishui. In 2000, the high demand for BGI showed the distribution characteristics of the primary and secondary cores. The primary core is located in an area of 7 km in length from north to south and 5 km wide from east to west. It is centered on Hunan Road Street. The primary core has the greatest overall social and economic demand. The secondary core is located in the Getang and the Dachang sub-cities, with increased social, economic, and environmental demands due to the development of heavy industry. The total demand at the junction of Jiangning District and the central urban area is usually larger than that of other areas in the territory (except Lukou Street) where location conditions are limiting factors. In 2010, the areas with high demand were largely reduced. The primary core was transferred to the downtown area centered on Yijiangmen Street, and the secondary core was moved to Nanjing Chemical Industry Park. Demand in Nanjing high-tech Development Zone, Lukou airports, and other areas has further increased. The demand along the southern bank of the Yangtze River upstream (Nanjing section) is steadily growing due to the continuous combination development of Binjiang new city, Banqiao new city, and Meishan Street. In recent years, Jiangbei main city, sub-city, new city, and new village have been the economic pioneers of rapid development. The old city’s ecological environment greatly improved, and demand was further reduced in 2020. However, the aggregate demand increased steadily in Jiangbei new city central, Luhe sub-city, Lishui sub-city, Gaochun sub-city, Nanjing high-speed railway hub zone, and Hexi CBD regions, where the rapid development of industry and population aggregation is taking place.

### 3.2. Influencing Factors of Blue–Green Infrastructure Demand

The results of the regression analysis are presented in Table 4 below. The R^2^ of the model is 0.5642, indicating that the independent variable explains 56.42% of the BGI demand distribution. In addition, the F-statistic of the model was 2274.80, with a *p*-value less than 0.001, indicating a good model effect.

Overall, the level of economic development, urban spatial layout, and decision management orientation all positively influence the BGI demand distribution. Among them, the urban spatial layout has the greatest impact on BGI demand (0.501), followed by GDP (0.296) and new parks (0.120), and the industrial park distribution has the least impact (0.051).

#### 3.2.1. Level of Economic Development

The level of economic development is a crucial factor that determines the demand for BGI. Nanjing is considered the core city of the Shanghai–Ningbo–Hangzhou regional manufacturing base; its industrial-added value accounted for more than 40% before 2015. Industrial development always has high requirements for service functions, such as air purification, carbon sequestration services, temperature regulation, etc.

On the one hand, industrialization promotes the improvement of the regional economic performance (Table 5). The larger the industrial added value is, the greater the fixed asset investment and land development intensity will be. Further, there will be an increase in the demand for BGI supply and production services. On the other hand, areas with favorable economic conditions can attract highly educated and high-quality talents, which increases the number of permanent urban residents. As urban residents’ disposable incomes rise, they are more inclined to improve the quality of their working and living environments. This increases their demand for social, ecological, and environmental BGI. In addition, the area with high BGI demand overlaps with the manufacturing park to a certain extent (Figure 7), indicating that the agglomeration development of the manufacturing industry requires BGI to provide the corresponding service functions. In addition, the change in fixed asset investment patterns indirectly affects BGI demand for regional economic development through its impact on manufacturing, real estate, and other industries.

#### 3.2.2. Urban Spatial Pattern

The urban spatial pattern has the greatest impact on the distribution of BGI demand. The urban spatial layout determines to a certain extent the distribution of urban population, resources, transportation, and infrastructure, which in turn affects the distribution of BGI demand. Nanjing gradually formed an urban spatial pattern of “two main cities, three sub-cities, and nine new cities” (Figure 8). Jiangnan’s main city has become the core region of a world-famous cultural town, a highland of advanced technology industries, and the densest population area, which is also the highest economic, social, and ecological demand area in Nanjing. Results are consistent with the description of high BGI demand concentrated in the central city area. This pattern was due to downtown Nanjing’s outward expansion, the manufacturing area shifting to the city periphery, and the refined urban function. Jiangbei’s new city is viewed as the core area of the national demonstration new area. This area carries the intersection function of the Yangtze River Economic Belt and the Yangtze River Delta urban agglomeration. The manufacturing industry is highly developed, and the demand for BGI in the region is generally higher than that in the surrounding areas. Luhe sub-city, Lishui sub-city, and Gaochun sub-city are the suburbs where the population and industry are gathering, and the demand for BGI is much larger than that of the surrounding areas. Lukou, Zetang, Longtan, Longpao, Qiaolin, Binjiang, Banqiao, Tangshan, and Chunhua are nine new towns implementing specialized labor divisions. The nine new towns act as a strategic aggregation of emerging industries, and the demand for BGI is relatively high. In conclusion, the spatial layout of the main city, sub-city, and new city in Nanjing is consistent with the distribution pattern of BGI demand. The urban spatial layout greatly impacts the distribution of BGI demand.

#### 3.2.3. Decision Management Orientation

Government departments’ decision-making and management orientation play a crucial role in developing BGI demand. The construction of eco-civilization, the continuous promotion of Beautiful China, the conservation of the Yangtze River, and the implementation of the “Green Nanjing” strategy all show the serious attention that the Nanjing Municipal government attaches to the protection and construction of urban eco-environment, the dependence and coordination of urban green space system planning, and overall urban planning. In 2004, the “Opinions of Nanjing Municipal Government on the Implementation of the ‘Green Nanjing’ Strategy to Build an Ecological City” proposed the idea of “supporting the growth of BGI in the main urban area and some suburbs.” In 2014, the “Ecological Red Line Area Protection Plan of Nanjing” reserved 24.75% of the land area as the protected red line and ecological management area. The plan provides an invaluable opportunity for BGI construction. In 2019, the “Nanjing Ecological Civilization Construction Plan 2018–2020 (Revised)” was formulated to actively encourage comprehensive environmental improvement and ecological restoration with the ecological red line as the bottom line. “Nanjing 14th Five-Year Ecological and Environmental Protection Plan” proposed to enhance ecosystem services, promote the comprehensive management of “mountains, forests, fields, lakes, grass, and sand,” and systematically solve the development bottleneck of BGI in 2021. Continuous investments in human resources, material, and financial resources maintain the balance between supply and demand for BGI. Environmental investments promote the gradual increase in urban residents’ demand for improved living and working environments and are an effective guarantee for the virtuous development of BGI. An imperative role for decision management orientation is to guide the alteration of BGI’s spatial and temporal patterns.

## 4. Discussion

### 4.1. Comprehensive Blue–Green Infrastructure Demand Evaluation System

BGI is a green space network that plays a significant role in the process of city construction. Its construction is meant to meet the increasing economic, social, environmental, and ecological demands of urban development, and it is an effective guarantee of sustainable urban development [45].

There is no recognized index system or accounting method for BGI demand evaluation, which is different from supply evaluation. Existing demand evaluation systems tend to overlook certain aspects of residents’ demand for BGI. Some of evaluation frameworks appear to be anthropocentrism and utilitarianism, and their environmental and ecological indicators are often ignored and underestimated [46]. In order to comprehensively consider and scientifically estimate the demand for urban construction for the diversified service functions of BGI, this study argues that the evaluation system of BGI demand should integrate and couple the four dimensions of social, economic, ecological, and environmental research. BGI has significant air purification and temperature regulation functions, which can provide environmental services for urban residents [47]. At the same time, ecosystem integrity significantly impacts human health and well-being, and climate change is an imperative issue for human destiny [48]. BGI is an instrumental component of ecosystem carbon sinks based on natural solutions and is also a key element in ecosystem network construction [35,49]. However, this indicator system still has some limitations. Since there are different ecosystem service values between blue infrastructure and green infrastructure, there may be differences between the demand of residents for blue infrastructure and green infrastructure. This indicator system does not distinguish the difference in residents’ demand for blue–green infrastructure, which still needs to be improved in the subsequent studies.

### 4.2. Layout of Blue–Green Infrastructure Demand for Mega Cities

Nanjing, as a mega-city in China, is an appropriate area to study the distribution of BGI demand due to the concentration of population and industry. BGI demand in Nanjing is consistent with the changing trend of urban spatial patterns. This trend shows a gradually decreasing distribution pattern from the core area to the inner and outer suburbs. Among them, the city core area is the region with high demand for BGI due to its characteristic as the center of human activities [50,51]. Taking into account time change, the BGI demand in the core area and the outer suburbs gradually decreased, and the inner suburbs first decreased and then increased. The core area is the main driving force of early social and economic development. Residents’ demand for BGI tends to saturate as the economy and social development approach saturation. In addition, the population has gradually moved out, and the suburbs have become the focus of urban development and construction; with the development of cities, the demand for BGI has increased yearly. However, the degree of population and industrial agglomeration in the outer suburbs has decreased compared with the core area and the inner suburbs. This is because they are limited by the level of economic development.

Through other studies of BGI demand in mega-cities, such as Zhengzhou and Barcelona, areas with high BGI demand tend to be in urbanized areas [14,18]. The first thing that BGI should address in mega-cities is urbanized area renovation. These places have neglected the construction of BGI due to early development, and urban renewal and renovation should first consider the optimization of BGI in this area. However, often these areas have high building density, and it is appropriate to adopt strategies for local renovation, such as pocket parks and green roofs. Suburban areas are often urbanizing areas, and these areas need to be planned with a long-term perspective weighing the layout of construction land and BGI to avoid the status quo of urbanizing areas. The outer suburbs, on the other hand, should optimize the structure of BGI and optimize the quality of the existing stock.

### 4.3. Influencing Factors of Blue–Green Infrastructure Demand

Urban residents have a large demand for BGI, so the factors affecting their demand are inextricably linked to human activities. In our study, we found that the urban spatial pattern has the greatest impact on the BGI demand distribution. The intra-city hierarchy largely determines the layout of urban industries, population, transportation, infrastructure, etc. [52]. The main city tends to be the earliest developing area of the city, with concentrated industries and frequent human activities, and has a greater demand for BGI; the secondary city and new city are the key areas of the city’s current development and have a higher demand for BGI. The level of economic development determines the demand for BGI. The level of regional economic development is closely related to various factors, including fixed asset investment and land development intensity. In addition, well developing/developed regions are more likely to attract high-quality talents, and high-quality talents have a high demand for delicate ecology and environment [53]. The production activities of industrial enterprises cause environmental pollution and ecological system damage, so industrial enterprises mainly increase ecological and environmental demands for BGI. On the contrary, there are corresponding demands for social and economic aspects when more workers gather in industrial enterprises. In addition, government planning, construction, decision-making, and management of BGI not only directly affect the demand for BGI but also solve the spatial inequality of supply and demand of BGI to a certain extent and achieve environmental justice [54].

However, this study still has some limitations. BGI demand measurements are mainly based on four dimensions of society, economy, environment, and ecology. The accuracy of measurement results is affected by the representativeness of selected indicators, data availability, etc. The absolute quantity does not affect the study of spatial–temporal pattern changes in the same region. The optimization of BGI patterns requires accurate accounting according to research units, so understanding the relative representativeness of the indicators is also crucial. In addition, the lack of comprehensive factors determines that various factors’ effects on BGI demand variations cannot be quantitatively analyzed in one-to-one correspondence, but can only be qualitatively analyzed using the spatial growth and decline trend. Some studies have begun to notice the neglected BGI in informal areas, low-income communities, pointing out the barriers to multifunctional BGI in these areas, arguing that factors, such as identity, local attachment, and community culture, also contribute to BGI development [55]. These have been overlooked in studies of BGI in mega-cities.

## 5. Conclusions

This study builds a BGI demand evaluation system based on four dimensions: social, economic, environmental, and ecological, and analyzes the temporal and spatial results of BGI demand in Nanjing. On this basis, the factors affecting the demand change for BGI are analyzed, and the following conclusions are obtained:

The BGI social demand in Nanjing first increased and then decreased from the year 2000 to 2020, and the high demand was mainly concentrated in the downtown area. BGI economic demand decreased as a whole, and the main urban area’s economic demand decreased as well. On the contrary, the economic demand for the suburbs and the outer suburbs kept growing. The ecological demand for BGI was increasing yearly, and the spatial pattern shows that high ecological demand was scattered around the central city on the southern side of the Yangtze River and the new central city on the northern side of the Yangtze River. The environmental demand for BGI was decreasing. The high environmental demand areas were concentrated in Luhe District, the main urban area, and the northern region of Jiangning District. The environmental demands of Lishui District and Gaochun district were decreasing significantly.

Changes in the BGI demand are mainly influenced by human activities. The urban spatial pattern has the greatest impact on the distribution of BGI demand. In the process of urban development, the economic development level directly affects the economic demand for BGI. The adverse effects of industrial development directly increase demand while, on the other hand, urban development attracts people and improves their living standards, indirectly increasing demand. The spatial layout of the central city, sub-city, and new city in Nanjing is consistent with the spatial distribution pattern of BGI demand. Cities’ construction and industrial development significantly impact the variations in demand patterns. Since 2000, the Nanjing government’s departments have taken various measures, such as introducing ecological and environmental aspects in planning and investing in ecological and environmental management funds, to guide the market and society and other themes to focus on ecological and environmental issues, to maintain the balance between the supply and demand of BGI and directly promote continuous BGI improvement based on decision-making and management.

## Figures and Tables

**Figure 1 ijerph-20-03979-f001:**
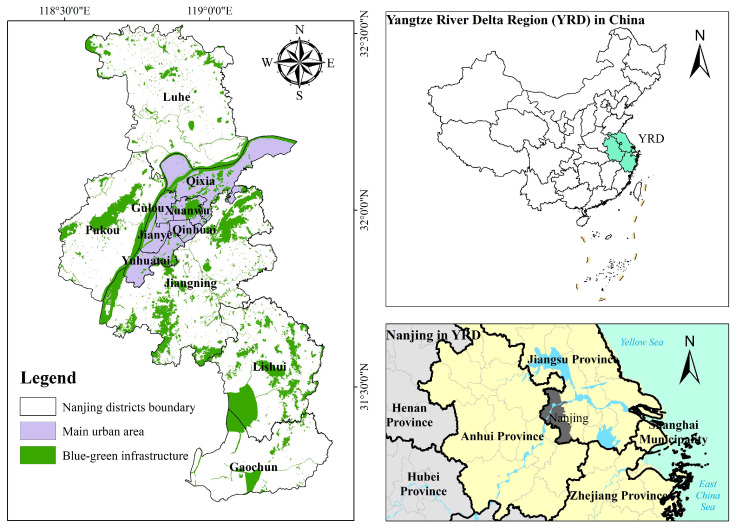
Nanjing’s location.

**Figure 2 ijerph-20-03979-f002:**
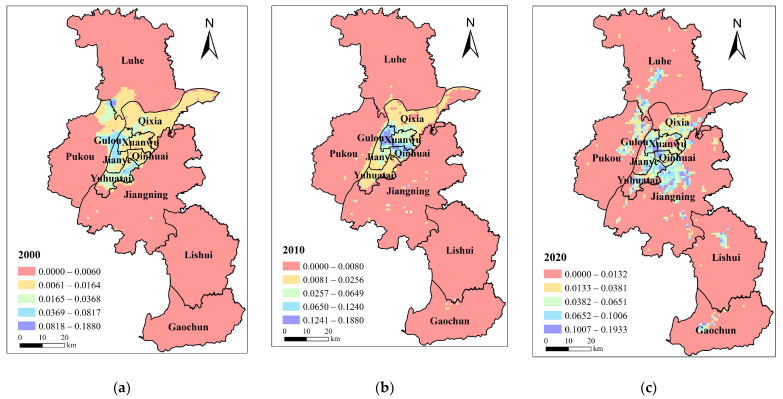
Social demand distribution pattern of blue–green infrastructure in Nanjing in (**a**) 2000; (**b**) 2010; (**c**) 2020.

**Figure 3 ijerph-20-03979-f003:**
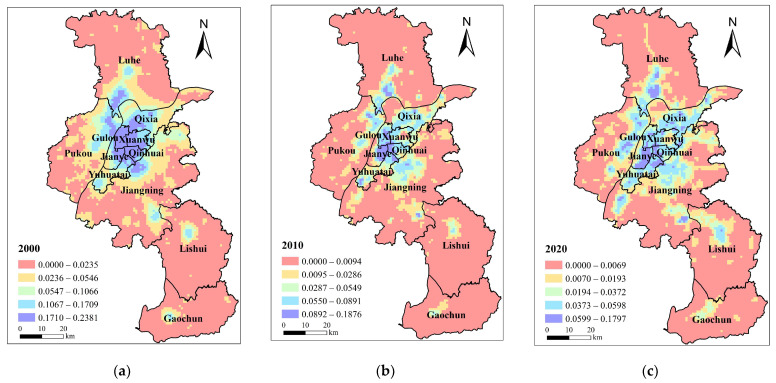
Economic demand distribution pattern of blue–green infrastructure in Nanjing in (**a**) 2000; (**b**) 2010; (**c**) 2020.

**Figure 4 ijerph-20-03979-f004:**
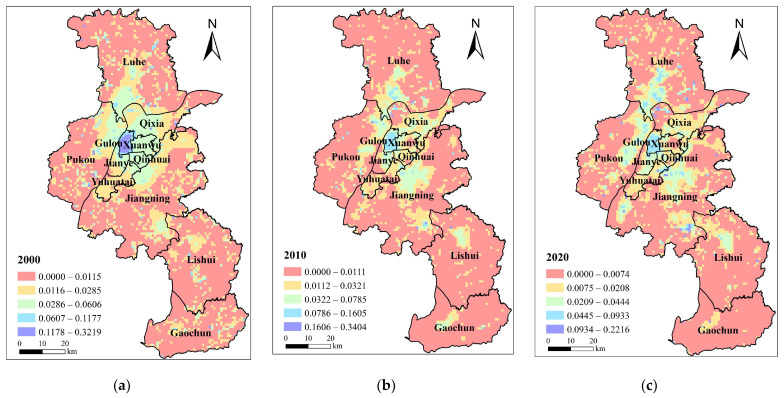
Ecological demand distribution pattern of blue–green infrastructure in Nanjing in (**a**) 2000; (**b**) 2010; (**c**) 2020.

**Figure 5 ijerph-20-03979-f005:**
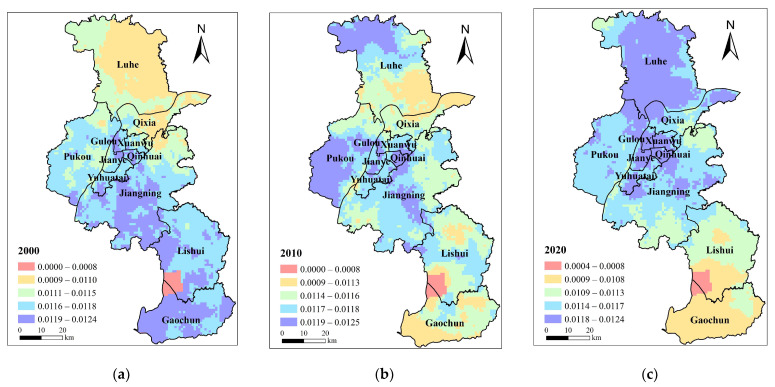
Environmental demand distribution pattern of blue–green infrastructure in Nanjing in (**a**) 2000; (**b**) 2010; (**c**) 2020.

**Figure 6 ijerph-20-03979-f006:**
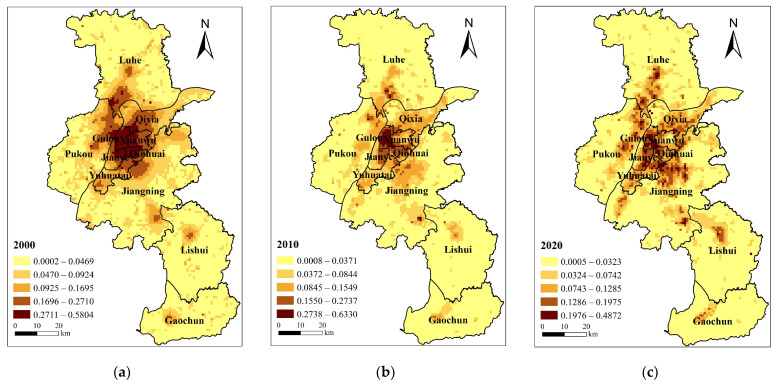
Overall demand distribution pattern of blue–green infrastructure in Nanjing in (**a**) 2000; (**b**) 2010; (**c**) 2020.

**Figure 7 ijerph-20-03979-f007:**
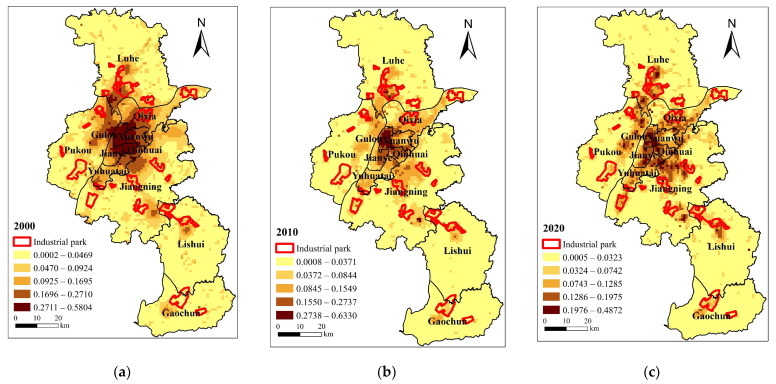
Layout characteristics of industrial parks and blue–green infrastructure demand in Nanjing in (**a**) 2000; (**b**) 2010; (**c**) 2020.

**Figure 8 ijerph-20-03979-f008:**
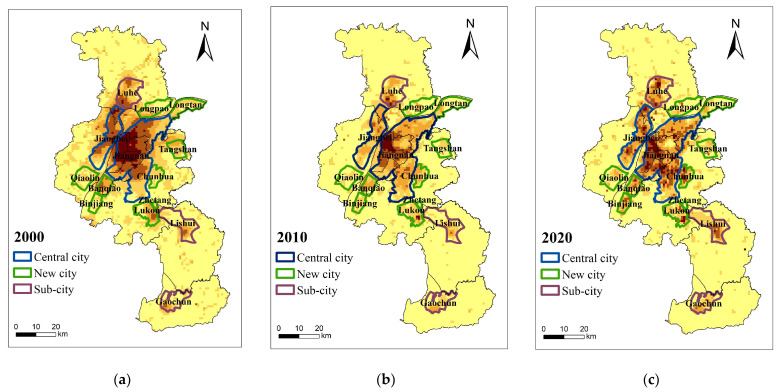
Urban spatial pattern and blue–green infrastructure demand of Nanjing in (**a**) 2000; (**b**) 2010; (**c**) 2020.

**Table 1 ijerph-20-03979-t001:** Carbon emission factors of various energy sources.

Type	Carbon Emission Factor
Coal	0.7329
Oil	0.5574
Natural gas	0.4226

**Table 2 ijerph-20-03979-t002:** Indicator weight.

System	Indicators	Information Entropy Value (*e*)	Information Utility Value (*d*)	Weight Factor (*w*)
Society	Population density	0.9412	0.0588	18.80%
	Building density	0.9541	0.0459	14.66%
Economy	Night light	0.9448	0.0552	17.65%
	Intensity of land development	0.9738	0.0262	8.37%
Ecology	Carbon emissions	0.9461	0.0539	17.24%
	Fragmentation	0.9311	0.0689	22.02%
Environment	Surface temperature	0.9997	0.0003	0.09%
	PM_2.5_	0.9963	0.0037	1.17%

**Table 3 ijerph-20-03979-t003:** The dependent and independent variables for regression analysis.

Factors	Variables	Instructions
/	*Demand*	The dependent variable
Level of economic development	*GDP*	Data are from the Resource and Environmental Science and Data Center (https://www.resdc.cn/, accessed on 28 March 2022).
	*Industrial*	According to whether the plot is an industrial park site or not, the values of 1 and 0 are assigned, respectively.
Urban spatial layout	*Function*	Plots are assigned values according to different classes of cities.
Decision management orientation	*Newpark*	Assign values of 1 and 0, respectively, according to whether there is an additional park on the plot.

**Table 4 ijerph-20-03979-t004:** The results of regression analysis.

Variables	Coef.	Std. Err.	t	$p\;>\;\left|t\right|$	Beta
*Industrial*	0.0079817	0.0012813	6.23	0.000	0.051056
*GDP*	2.72 $\times 10^{-7}$	8.44 $\times 10^{-9}$	32.2	0.000	0.296364
*Function*	0.0208163	0.0003914	53.19	0.000	0.500803
*Newpark*	0.0158649	0.0011213	14.15	0.000	0.120005
*Constant*	0.0165263	0.0004359	37.91	0.000	/

**Table 5 ijerph-20-03979-t005:** Economic development of Nanjing.

Administrative District	GDP/Billion CNY			Fixed Asset Investment/Billion CNY		
	2000	2010	2020	2000	2010	2020
Xuanwu	12.01	372.82	1108.66	13.53	91.6	152.28
Qinhuai	20.73	466.32	1286.6	20.79	162.41	247.61
Jianye	8.32	237.96	1121.53	13.91	175.89	460.54
Gulou	15.47	608.88	1772.6	25.08	185.54	297.06
Pukou	38.86	369.1	1407.06	12.38	410	903.68
Qixia	15.52	681.11	1569.15	14.01	290.84	552.23
Yuhuatai	29.77	214.87	947.14	18.16	214.18	318.04
Jiangning	101.34	678.58	2509.32	36.66	630	807.8
Luhe	49.68	575.99	1654.88	14.69	455.04	627.69
LiShui	38.27	250.16	911.51	10.53	224.53	383.56
Gaochun	37.33	247.26	513.13	9.8	190.11	211.77

## Data Availability

The data presented in this study are available on request from the corresponding author.

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
