# Peer review of "Socioeconomic Factors Influence the Spatial and Temporal Distribution of Blue–Green Infrastructure Demand: A Case of Nanjing City"

_ijerph, 2023, doi:10.3390/ijerph20053979_

Round 1

Reviewer 1 Report (Previous Reviewer 3)

The article is excellent. It provides an in-depth analysis of the socioeconomic factors of the area studied. For this purpose, the authors use a very current methodology which, in addition, is obtaining very interesting results such as the one presented here. On the other hand, the authors adequately contextualize the research and show the research carried out in a clear and understandable way. In addition, the incorporation of the maps allows us to understand the change over time. The discussion is extensive and relates the data to results obtained elsewhere.

I would simply suggest to the authors that on page 12 they insert spaces before Table 3, so that it can be viewed in its entirety on page 13. Once this has been done, try not to leave the title of the Discussion section isolated on page 14.

An excellent job, no doubt about it. Congratulations to the authors!

Author Response

Thank you for your acknowledgement of this study! In revision, we are making adjustments to the overall layout of the article, and we appreciate your suggestions!

Reviewer 2 Report (Previous Reviewer 2)

Since my earlier comments have been addressed, I do not have additional comments this time. I recommend it to go to the next step. 

Author Response

Thank you for your previous valuable comments on this article, and good luck with your work!

Reviewer 3 Report (New Reviewer)

This is an interesting case study that provides a comprehensive evaluation of the demand for blue-green infrastructure to support the rational distribution of blue-green infrastructure in urban areas. The study measures the demand for blue-green infrastructure in four aspects: society, economy, ecology, and environment, and the selection of indicators is generally reasonable and innovative. The conclusions of the article can better fit the actual situation in Nanjing. In general, I think this paper can be accepted by your journal. Also attached are some issues that need to be revised.

Line 35: built landscape should be used the term of natural landscape;

Line 41: people's need, changed as needs (not only for people, but also the nature)

Line 46: continued replaced by continuing is better

Line 68: construct may corrected as constructed

Line 86: please delete “an”

Line 100: Nanjing city is used as, should called as Nanjing, capital city of Jiangsu Province, China is studied as

Line 102: It also, should changed as We also

Line 121: Figure 1. Nanjing's location and its blue-green infrastructure distribution, we can not understand what is so-called “its blue-green infrastructure distribution. Figure 1 makes no sense.

Line 183: Why does the author in the figure center use only “Green infrastructure benefits” rather than blue-green infrastructure

Line 261: the word “presents” replaces as shows

Line 303: through Line 443, the captions of all Figures 3,4,5,6,7 should give the meanings of the different colors as well as /represented by the numerical values ranges)

Line 549: the expression of “high levels of economic development regions” instead of “ well developing/developed regions”

Line 578: The serial number (1) is neither necessary nor reasonable because you have only 2 conclusions listed. Similarly, Line 589: The serial number (2) should be deleted.

Author Response

Thank you for your detailed comments on this paper, which will help a lot to improve and enhance this paper. In this revision, we have made changes based on your comments, and the tracking mode has been turned on in the revision.

Reviewer 4 Report (New Reviewer)

Dear Authors,
although the presented topic is interesting, in my opinion, your paper does not meet the requirements of a reliable scientific article.

Main objections:

(1)    Introduction: This section needs strong improvement. Authors should clearly indicate research gap, state the research questions and hypothesis. In addition, Autors should clearly indicate the novelty of their research. Now these elements are missing.
I noticed that in line 99 the authors are trying to give a research gap, writing that (…) the distribiution of demand for BGI is mainly influance of socioeconomic development, but relevant studies are still relatively limited, but it is not enouhgt, especially since there is no in-depth consideration or use of more elaboratec socio-economic indicators in the paper.

(2)    Materials and Methods:
Section 2.2. needs significant improvement. For this section, I expect a strong reliance on scientific sources, which is now lacking. This section should precisely refer to the selected variables that build the final index (overall/total demand). The Authors should provide not only general information, but a thorough explanation of the choice of each index-forming factor over other possibilities. The selection of indicators for the "ecological" dimension is suprising for me. The Authors should consider differences between ‘ecological’ and ‘environmental’, which are inadequately addressed here.

(3)    Figure 2 may confuse. I don't know if the purpose of Figure 2 was to present the benefits of BGI in general, or the Autors want to refer to the proposed indicators - which is suggested by the division into 4 sections. I believe that the chosen factors do not fully capture the benefits of BGI. I suggest either removing the figure or strongly supplementing /add more benefits presented in the four blocks.
If the Autors  wanted to refer to the used indicators there is a lack of consistency: in Figure 2, in the 'ecological' section, animal habitat is mentioned, but in the text (in lines 159-169) there is no detailed discussion of this issue. On the other hand, in Table 2 (line 259) space fragmentation appears - as an indicator referring to animal habitat? - this is a huge oversimplification, especially since the used indicator (fragmentation index) says nothing about the continuity of BGI.
Selection, description and justification, as well as assignment of a given indicator to one of the four sections are the weakest part of the work and must be improved/ better explained based on the literature.

(4)     Building density, night lights and intensity of land development contain very similar information.  For this reason, in my opinion, the authors should rethink whether all of the above variables should be selected for the final indicator (total/ overall demand)

(5)    There is nothing in the theoretical part about BGI fragmentation. Please complete your work with information on the impact of fragmentation of landscape/BGI on ecosytems or/and BGI demand.

(6)    Section 2.2.2. should be significantly expanded. Entropy method should be more explained in this section. 

(7)    Information from section 2.3. after completing the literature references, should be merged with the introduction section.

(8)    Discussionsection is weak. Especially the contextualization of findings within pevious research is missing.

Author Response

Thank you for your detailed and valuable comments on this article, which helped a lot to improve it. Based on your suggestions, we have made detailed revisions to the article.
(1) We have made major revisions on the introduction section to emphasize the research gaps and the contribution of this paper.
(2) Materials and Methods: 1) Literature citation was added to section 2.2, and specific explanation of selected indicators was added; 2) Figure 2 was deleted to avoid unnecessary ambiguity; 3) Explanation of entropy method was added to section 2.2.2; 4) Section 2.3 was added to the introduction section, literature citations were added, and quantitative methods were added to determine the different effects of influencing factors on BGI demand.

(3) Discussion section: literature citation was added.

Round 2

Reviewer 4 Report (New Reviewer)

Dear Authors,

thank you for completing the article and responding to my comments. I believe that the article has gained a new quality. I approve the article for publication.

Good luck

This manuscript is a resubmission of an earlier submission. The following is a list of the peer review reports and author responses from that submission.

Round 1

Reviewer 1 Report

First of all, I'm not convinced by this study. Numerous studies have been conducted on demand for green infrastructure, particularly those that value ecosystem services. Many of these valuations of ecosystem services necessitate considering how much residents are willing to pay for the ecological and environmental regulation services that green spaces provide (such as the cooling and oxygen release mentioned by the authors). Therefore, the authors' claim that the public has been economically neglected in relation to the ecological components of existing studies is untrue. Moreover, existing studies on the demand for green infrastructure do not take into account ecological and environmental indicators (at least indicators such as pollution, cooling, etc., as in this paper), for example, Assessing the ecological balance between supply and demand of blue-green infrastructure, Negotiating value and priorities: evaluating the demands of green infrastructure development, so it is difficult to understand the authors' claim that environmental and ecological indicators in the evaluation of demand for existing green infrastructure It is therefore difficult to understand the authors' claim that environmental-ecological indicators are neglected and underestimated in the evaluation of existing green infrastructure needs.

Second, the four indicators' dimensions are extremely perplexing. First of all, I am unaware of the criteria used to divide these four dimensions. In fact, the indicators used in 2.2 and figure 2 differ from one another. For instance, in 2.2, there is no indicator for recreation in the economy or education in society, while landscaping can also be included under the environmental and ecological dimensions. The categorization of the indicators in the four dimensions of 2.2 feels confusing, for example, the carbon emissions indicator in the ecological dimension can be placed in the economic and environmental dimensions. The temperature of the ecological dimension can be placed in the environmental dimension, and even the fragmentation of landscape patterns as an important element of landscape ecology research can be placed in the ecological dimension. And many of the indicators chosen are in trade-off and coordination with each other, which the authors have not considered, and now all so cause the four results in the context of the whole four dimensions to be unconvincing.

Thirdly, the authors note that the way the four dimensions are now presented is not compelling and that numerous influencing elements, such as GDP, are the indicators that should really be studied by the four dimensions above.

Fourthly, the whole discussion is rather weak and it feels that neither the contribution of the article is exalted nor the results are well explained and compared with the existing green infrastructure needs and impacts made by other cities, we do not see the difference between the needs of the case and the green infrastructure needs of Barcelona or Wuhan, etc. in the current study.

Author Response

Dear reviewer 1,

Thank you for your valuable and detailed comments for the enhancement of this paper. Here are a few responses to our revision of this paper.

Your comments: First of all, I'm not convinced by this study. Numerous studies have been conducted on demand for green infrastructure, particularly those that value ecosystem services. Many of these valuations of ecosystem services necessitate considering how much residents are willing to pay for the ecological and environmental regulation services that green spaces provide (such as the cooling and oxygen release mentioned by the authors). Therefore, the authors' claim that the public has been economically neglected in relation to the ecological components of existing studies is untrue. Moreover, existing studies on the demand for green infrastructure do not take into account ecological and environmental indicators (at least indicators such as pollution, cooling, etc., as in this paper), for example, Assessing the ecological balance between supply and demand of blue-green infrastructure, Negotiating value and priorities: evaluating the demands of green infrastructure development, so it is difficult to understand the authors' claim that environmental and ecological indicators in the evaluation of demand for existing green infrastructure It is therefore difficult to understand the authors' claim that environmental-ecological indicators are neglected and underestimated in the evaluation of existing green infrastructure needs.

Our response: In the introduction section, we have elaborated on the measurement of GI demand in general into three approaches: (i) evaluation system based on ecosystem services; (ii) evaluation system by indicators construction; (iii) questionnaire survey on willingness to pay, etc. Some studies have been conducted by surveying residents' willingness to pay and the number of payments for urban green spaces (in reference [16]), and also elaborated on the measurement of GI demand based on ecosystem services (e.g., the paper Assessing the ecological balance between supply and demand of blue- green infrastructure, reference [12]). These studies include economic, social, ecological and environmental aspects of GI demand. However, there is still little literature that considers the above four aspects in an integrated manner, and this study attempts to construct a GI demand evaluation system from these four aspects. It may be that some of the statements in the introduction section may cause ambiguity to the reviewers, and we add them in the introduction section.

Your comments: Second, the four indicators' dimensions are extremely perplexing. First of all, I am unaware of the criteria used to divide these four dimensions. In fact, the indicators used in 2.2 and figure 2 differ from one another. For instance, in 2.2, there is no indicator for recreation in the economy or education in society, while landscaping can also be included under the environmental and ecological dimensions. The categorization of the indicators in the four dimensions of 2.2 feels confusing, for example, the carbon emissions indicator in the ecological dimension can be placed in the economic and environmental dimensions. The temperature of the ecological dimension can be placed in the environmental dimension, and even the fragmentation of landscape patterns as an important element of landscape ecology research can be placed in the ecological dimension. And many of the indicators chosen are in trade-off and coordination with each other, which the authors have not considered, and now all so cause the four results in the context of the whole four dimensions to be unconvincing.

Our response: Figure 2 represents the benefits of green infrastructure from four dimensions, and based on the four benefits we choose the indicator of 2.2. In the social dimension, we mainly consider residents' health and well-being and landscape appreciation. People are the main body of the society and the social demand of GI mainly comes from the residents, so the two indicators of population density and building density are considered. In the economic dimension, development and ecosystem maintenance are taken into account. The economic dimension is distinguished from the social dimension by considering mainly development issues. On the one hand, concepts such as ecological economy and green economy are injected into economic development, and on the other hand, rough urban development is no longer what sustainable urban development should look like, and green infrastructure contributes to high-quality urban development. Therefore, two indicators, night lighting and urban development intensity, are considered. In the ecological dimension, we mainly consider the carbon sequestration function and habitat maintenance of green infrastructure. True to the reviewer's suggestion, the landscape fragmentation indicator should be in the ecological dimension. The carbon emission indicator characterizes the need for carbon sequestration and also provides the basis for urban carbon neutrality, so the carbon emission indicator is placed in the ecological dimension rather than the economic or environmental indicators. In the environmental dimension, climate and pollution regulation are considered. Based on the city-scale study, we consider the green infrastructure in mitigating the heat island effect, and therefore the surface temperature is selected. And pollution is considered mainly considering PM2.5 as an important air pollutant.

Your comments: Thirdly, the authors note that the way the four dimensions are now presented is not compelling and that numerous influencing elements, such as GDP, are the indicators that should really be studied by the four dimensions above.

Our response: When we selected the indicators, we first considered whether the indicators could characterize the four dimensions, and secondly, the availability of data. The specific selection of indicators has been described in the previous response. As for the GDP mentioned by the reviewer, in fact, a large amount of literature (such as [26][27]) points out that there is a covariance between night lighting data and GDP, so in this study, night lighting was selected to characterize the economic dimensions.

Your comments: Fourthly, the whole discussion is rather weak and it feels that neither the contribution of the article is exalted nor the results are well explained and compared with the existing green infrastructure needs and impacts made by other cities, we do not see the difference between the needs of the case and the green infrastructure needs of Barcelona or Wuhan, etc. in the current study.

Our response: The discussion section is supplemented with an emphasis on adding the differences between this paper's research and other studies, adding how to optimize the layout of green infrastructure in mega-cities based on the results, and adding the latest literature.

Reviewer 2 Report

This is an interesting case study about what comprehensive assessment for green infrastructure in an urban built environment is like and how to carry out such a comprehensive assessment strategy. The authors best characterize the four dimensions of green infrastructure demands. The carefully chosen indicators of each dimension of green infrastructure demands are calculated correctly. All GIS output maps are of good quality and quite effectively communicate the analysis results. If interviews with the key city planning figures and/or conducting focus groups, more data about why the green infrastructure demands are high in the center and low at the peripheral locations would be better understood. Of course, that is not the scope of this research. In the Conclusion section, I would like to see a little bit of discussion on what the Nanjing government specifically does to balance the supply and demand of green infrastructure. 

Author Response

Dear Reviewer 2,

Thank you for your pertinent comments on this paper. In response to your comments, we have made the following improvements.

Your comments: If interviews with the key city planning figures and/or conducting focus groups, more data about why the green infrastructure demands are high in the center and low at the peripheral locations would be better understood. Of course, that is not the scope of this research.

Our Response: Yes, interviews can certainly provide a better explanation of green infrastructure demand. We have already conducted this in another study, which is not the issue addressed in this article. The interview is currently in progress. Thank you for your valuable comments.

Your comments: In the Conclusion section, I would like to see a little bit of discussion on what the Nanjing government specifically does to balance the supply and demand of green infrastructure.

Our Response: The policies adopted by Nanjing government in balancing the supply and demand for green infrastructure have been elaborated in section 3.2.3. We did not include it in the conclusion section because we considered that this section should not be too long. In this revision, we have added some concluding statements in the conclusion section.

Reviewer 3 Report

The paper is excellent. The evaluation system used is very appropriate. Furthermore, this paper is very well contextualised and well understood. The bibliography is up to date and appropriate to the objective of the research. I would simply like to ask the authors to use softer colours in figure 2 so that it can be seen better. Otherwise, I can only offer my sincere congratulations for the research carried out.

Author Response

Dear Reviewer 3,

Thank you very much for your positive comments on this paper, your confirmation is very encouraging for us. In response to your comments, we have made the following improvements.

Your comments: I would simply like to ask the authors to use softer colours in figure 2 so that it can be seen better.

Our Response: Have changed Figure 2 to a softer color.